# Peer review of "Consensus-Based Guidelines for Best Practices in the Selection and Use of Examination Gloves in Healthcare Settings"

_nursrep, 2025, doi:10.3390/nursrep15010009_

Round 1
Reviewer 1 Report
Comments and Suggestions for Authors
The article discusses a topic of great interest to healthcare professionals; however, some adjustments are suggested for its publication.
Introduction
1. Include WHO reference on SDGs at the end of the sentence, line 80.
2. The introduction should be more robust. Include four more references to support the relevance of the topic.
3. The last paragraph of the introduction (lines 77 to 80) should be moved to the method chapter. Insert this paragraph as the first in the method chapter. State that the article followed the recommendations of the Accurate Consensus Reporting Document (ACCD), put it in full and then the acronym, stating that it is a reporting guideline for consensus methods in biomedicine developed via modified Delphi.
Method
Regarding ethical procedures, each country has its own legislation, but it may seem strange to readers from countries that require the requirement to go through the Research Ethics Committee (REC) whenever human beings are involved, regardless of whether or not there is an intervention. In Brazil, to conduct research with experts, that is, human beings, the project must go through the RCE, even if there is no intervention on the subjects. Therefore, I suggest including a sentence justifying that in the country of origin where the consensus was produced, it is not necessary to go through the RCE.
Conclusion
Rewrite the conclusion by returning to and responding to the proposed objective of the research, which is: “to provide detailed consensus-based guidelines on best practices for the selection, use, and disposal of examination gloves in healthcare settings”. Answer these questions: what are the best practices? When is their use indicated? Where should they be discarded? Summarize what was presented in the text in response to the objective. The conclusion can be more succinct, just two paragraphs, but it should respond to the objective of the research.
Author Response
Reviewer #1:
Thank you for taking the time to review again our manuscript and for your valuable feedback.
We appreciate the recognition regarding the improved version.
We have carefully considered your input.
- ”Include WHO reference on SDGs at the end of the sentence, line 80.”
Answer: We acknowledge your concern regarding the references and we added to specific ones about SDGs and the Health Workforce
We recognize the importance of using clear and succinct language to effectively convey our research findings. Please, see page 2, row 94.
Citations: [3]WHO (2017). World health statistics 2017: monitoring health for the SDGs, Sustainable Development Goals. Geneva: World Health Organization; 2017.
Licence: CC BY-NC-SA 3.0 IGO. | http://apps.who.int/iris/bitstream/10665/255336/1/9789241565486-eng.pdf?ua=1
[4]WHO (2016). Global strategy on human resources for health: workforce 2030. Geneva: World Health Organization; 2016. ISBN 978 92 4 151113 1. (http://apps.who.int/iris/bitstream/10665/250368/1/9789241511131-eng.pdf?ua=1)
- ” The introduction should be more robust. Include four more references to support the relevance of the topic.”
Answer: We have revised the introduction to make it more robust and have included four additional references to better support the relevance of the topic, as suggested. Please find the updated section in the revised manuscript. Please, see page 2 and 3, row 51-101.
- “The last paragraph of the introduction (lines 77 to 80) should be moved to the method chapter. Insert this paragraph as the first in the method chapter. State that the article followed the recommendations of the Accurate Consensus Reporting Document (ACCD), put it in full and then the acronym, stating that it is a reporting guideline for consensus methods in biomedicine developed via modified Delphi.”
Answer:Thank you for your suggestion. We have moved the last paragraph of the introduction (lines 77 to 80) to the beginning of the Methods chapter, as requested. Additionally, we have clarified that the article followed the recommendations of the Accurate Consensus Reporting Document (ACCD), specifying that it is a reporting guideline for consensus methods in biomedicine developed via the modified Delphi approach. This adjustment has been incorporated into the revised manuscript. Please, see page 3, row 111-115.
- “Method - Regarding ethical procedures, each country has its own legislation, but it may seem strange to readers from countries that require the requirement to go through the Research Ethics Committee (REC) whenever human beings are involved, regardless of whether or not there is an intervention. In Brazil, to conduct research with experts, that is, human beings, the project must go through the RCE, even if there is no intervention on the subjects. Therefore, I suggest including a sentence justifying that in the country of origin where the consensus was produced, it is not necessary to go through the RCE.”
Answer: Thank you for raising this important point. We have addressed your concern by including a sentence in the Methods section clarifying the ethical procedures followed. Specifically, we have stated that in the country of origin where this consensus was produced, it is not mandatory for projects involving expert consultations without interventions to undergo approval by a Research Ethics Committee (REC). This addition aims to ensure clarity for readers from countries where such approval is required for all research involving human participants. Please, see page 4, row 179-189.
- “Conclusion - Rewrite the conclusion by returning to and responding to the proposed objective of the research, which is: “to provide detailed consensus-based guidelines on best practices for the selection, use, and disposal of examination gloves in healthcare settings”. Answer these questions: what are the best practices? When is their use indicated? Where should they be discarded? Summarize what was presented in the text in response to the objective. The conclusion can be more succinct, just two paragraphs, but it should respond to the objective of the research.”
Answer: We have revised the conclusion to directly address the proposed objective of the research. The updated conclusion now succinctly summarises the best practices for the selection, use, and disposal of examination gloves in healthcare settings, highlighting key recommendations on their appropriate use and proper disposal as presented in the text. This ensures alignment with the research objective while providing a clear and focused overview conclusion. Please, see page 10, row 402-427.
Thank you for your thoughtful and constructive feedback. I have carefully addressed all the suggested revisions, and the manuscript has been updated accordingly.
Reviewer 2 Report
Comments and Suggestions for Authors
My basic concern is that all these guidelines are already available for healthcare professionals to use different types of gloves.. Why authors are trying to arrive at consensus? It is already known and approved
Authors have mentioned 30 experts, but not given their distribution with reference to their specialties, nations, expertise, etc. This distribution is required to ascertain the quality of consensus
I have another concern - Why round 2 was not done immediately after the in-person workshop? If you gave time, there is a definite possibility that people might have forgotten few/many things depending on the time spanned between round 1 and round 2.
Line 104-105 = It should be study (and not studies), as authors have cited one reference only
Author Response
Reviewer #2:
Thank you for taking the time to review again our manuscript and for your valuable feedback.
We appreciate the recognition regarding the improved version.
We have carefully considered your input.
- “Why are the authors trying to arrive at a consensus when guidelines are already available and approved for healthcare professionals?”
Answer: Thank you for your question. While guidelines for glove use exist, they often vary in scope, focus, and applicability, and are not always fully aligned with contemporary challenges in healthcare, such as sustainability and the One Health approach. This consensus document was developed to bridge gaps in existing guidelines by integrating international standards, emerging evidence, and practical insights from multidisciplinary experts. It aims to provide a more comprehensive, unified set of recommendations that address both clinical and environmental considerations, thus ensuring their relevance and applicability in diverse healthcare settings.
- “Authors have mentioned 30 experts, but not given their distribution with reference to their specialties, nations, expertise, etc. This distribution is required to ascertain the quality of consensus.”
Answer: Thank you for pointing this out. We have now included a texto summarizing the distribution of the 30 experts involved in the consensus process, detailing their specialties, and areas of expertise. This information highlights the diverse and multidisciplinary nature of the panel, which ensures that the consensus reflects a broad spectrum of knowledge and practical experience. Please, see page 3, row 116-131.
- “Why was Round 2 not conducted immediately after the in-person workshop? If time was given, there is a possibility participants might have forgotten some points.”
Answer: Thank you for raising this concern. The time between Round 1 and Round 2 was necessary to allow for the synthesis of the workshop's findings into a structured draft document for review. This period ensured that all recommendations were clearly articulated and provided participants with an opportunity to reflect on the discussions and provide more thoughtful feedback during Round 2. To mitigate the risk of participants forgetting key points, we shared detailed workshop notes and summaries, ensuring continuity and accuracy in the subsequent round of feedback. We include a paragrahp stating exacltly this calrification. Please, see page 4, row 154-162.
- “Line 104-105: It should be "study" (and not "studies"), as only one reference is cited.”
Answer: Thank you for identifying this oversight. The text has been corrected to "study" to align with the single reference cited. “The consensus threshold was set at 75% agreement, consistent with recommendations commonly found in Delphi methodology literature [11]”. We appreciate your attention to detail in this matter. Please, see page 4, row 151-152.
Thank you for your thoughtful and constructive feedback. I have carefully addressed all the suggested revisions, and the manuscript has been updated accordingly.
Reviewer 3 Report
Comments and Suggestions for Authors
Thanks for the opportunity to read this work.
Gloves, their selection and use, are clearly of interest for infection prevention.
This paper aims to present consensus-based guidelines. The method chapter corresponds to this aim, but I don't find these guidelines in the results chapter. To me, the results chapter shows a literature review. This is a major concern to me.
The abstract don't show any recommendations
I think the abstract and the results chapter should be rewritten to clearly present the guidelines. The references could also be improved.
Lines 75-76: United Nations reference should be added
Lines 82-88: are these guidelines nationally or internationally construct? Where do the experts come from? How representative are the experts? How were the experts' conflicts of interest managed?
Lines 212-213: some writing errors in the paper like health- care or env- vironments
Author Response
Reviewer #3:
Thank you for taking the time to review again our manuscript and for your valuable feedback.
We appreciate the recognition regarding the improved version.
We have carefully considered your input.
- The results chapter seems like a literature review rather than presenting guidelines.
Thank you for sharing your perspective on the results chapter.. We have revised the results chapter to clearly present the guidelines developed through the consensus process. We understand your concern and have taken steps to ensure the chapter clearly reflects the consensus-based guidelines rather than resembling a literature review. Our aim has been to present actionable recommendations derived from the consensus process, and we have made revisions to better highlight these guidelines and their practical implications for healthcare professionals. We greatly value your feedback and believe these updates strengthen the clarity and focus of the chapter.
- The abstract does not show any recommendations.
We appreciate your feedback. The abstract has been rewritten to include a summary of the key recommendations from the guidelines, emphasizing best practices for glove selection, usage, and disposal. This ensures that readers can quickly understand the practical contributions of the study. Please, see page 1, row 29-48.
- Line 75-76: United Nations reference should be added.
Thank you for pointing this out. We have added the appropriate United Nations reference to lines 75-76 to support the mention of the Sustainable Development Goals (SDGs). This citation provides credibility and context to the alignment of our guidelines with global sustainability frameworks. Please, see page 2, row 95.
- Are these guidelines nationally or internationally constructed? Where do the experts come from? How representative are the experts? How were the experts' conflicts of interest managed?
These guidelines were constructed with an international perspective, leveraging the expertise of 30 healthcare professionals from various specialties, including infection control, oncology nursing, wound care, perioperative nursing, and medical device standards. While the majority of experts are based in Portugal, their affiliations with international organizations such as the European Wound Management Association (EWMA) and global healthcare initiatives ensure a broad representation of perspectives.
To address potential conflicts of interest, all experts provided declarations of interest, and no conflicts were reported that could compromise the objectivity of the guidelines. This information has been added to the manuscript for transparency. Please, see page 3, row 116-131.
- Lines 212-213: Writing errors such as "health-care" and "env-vironment" need correction.
Answer: Thank you for identifying these errors. We have corrected "health-care" to "healthcare" and "env-vironment" to "environments" in the manuscript. Additional proofreading has been conducted to ensure there are no similar typographical errors.
Thank you for your thoughtful and constructive feedback. I have carefully addressed all the suggested revisions, and the manuscript has been updated accordingly.
Reviewer 4 Report
Comments and Suggestions for Authors
This manuscript presents a consensus-based approach to developing guidelines for examination glove use in healthcare settings. The study employs the Delphi method, involving a panel of 30 experts to establish evidence-based recommendations for glove selection, use, and disposal.
Strengths:
1. Methodology: The multi-stage Delphi process ensures a systematic and collaborative approach to developing guidelines, with a clear consensus-building mechanism.
2. Comprehensive Scope: The guidelines address critical aspects of glove use, including regulatory standards (EN 455), infection control, and sustainability considerations.
3. Interdisciplinary Approach: The panel includes experts from infection control, nursing, oncology, and wound care, providing a holistic perspective.
4. Integration of One Health and Sustainable Development Goals: The guidelines go beyond traditional medical considerations to incorporate broader environmental and sustainability perspectives.
Suggestion for Improvement: I recommend emphasizing the European context more explicitly in the title and throughout the text. Consider revising the title to: "European Consensus-Based Guidelines for Best Practices in the Selection and Use of Examination Gloves in Healthcare Settings". The document inherently possesses a strong European dimension by grounding its recommendations in key European regulatory frameworks, such as Regulation (EU) 2017/745 and the EN 455 standard series. The guidelines' emphasis on European production, with parameters that promote glove quality including a preference for European manufacturing, further underscores its European character. While the recommendations are deeply rooted in the European healthcare context and regulatory landscape, the comprehensive approach to infection control, sustainability, and professional best practices renders these guidelines potentially adaptable and instructive for healthcare systems in other geographical regions.
Overall, this is a significant contribution to healthcare infection control practices, offering a comprehensive and forward-thinking approach to examination glove use.
Author Response
Reviewer #4:
Thank you for taking the time to review again our manuscript and for your valuable feedback.
We appreciate the recognition regarding the improved version.
We have carefully considered your input.
- “Suggestion for Improvement: I recommend emphasizing the European context more explicitly in the title and throughout the text. Consider revising the title to: "European Consensus-Based Guidelines for Best Practices in the Selection and Use of Examination Gloves in Healthcare Settings". The document inherently possesses a strong European dimension by grounding its recommendations in key European regulatory frameworks, such as Regulation (EU) 2017/745 and the EN 455 standard series. The guidelines' emphasis on European production, with parameters that promote glove quality including a preference for European manufacturing, further underscores its European character. While the recommendations are deeply rooted in the European healthcare context and regulatory landscape, the comprehensive approach to infection control, sustainability, and professional best practices renders these guidelines potentially adaptable and instructive for healthcare systems in other geographical regions. Overall, this is a significant contribution to healthcare infection control practices, offering a comprehensive and forward-thinking approach to examination glove use.”
Answer: Thank you for your thoughtful suggestion regarding emphasizing the European context in the title and throughout the text. We agree that the guidelines inherently possess a strong European dimension, as they are grounded in key European regulatory frameworks such as Regulation (EU) 2017/745 and the EN 455 standard series. Additionally, the emphasis on European production and parameters that promote glove quality, including a preference for European manufacturing, further supports this perspective.
Throughout the text, we ensured that the European context is more explicitly highlighted by referencing the foundational regulatory and production standards that shaped these guidelines. At the same time, the document’s comprehensive approach to infection control, sustainability, and professional best practices will remain accessible and instructive for audiences worldwide. Please, see page 1, row 46-48 | page 3, row 99-110 | page 11, row 426-427
We appreciate your recognition of the document’s significant contribution to healthcare infection control practices and its forward-thinking approach to glove management.
Thank you for your thoughtful and constructive feedback. I have carefully addressed all the suggested revisions, and the manuscript has been updated accordingly.
Round 2
Reviewer 3 Report
Comments and Suggestions for Authors
The authors take into account all my recommendations.